# Diagnosis and Management of Seronegative Myasthenia Gravis: Lights and Shadows

**DOI:** 10.3390/brainsci13091286

**Published:** 2023-09-05

**Authors:** Claudia Vinciguerra, Liliana Bevilacqua, Antonino Lupica, Federica Ginanneschi, Giuseppe Piscosquito, Nicasio Rini, Alessandro Rossi, Paolo Barone, Filippo Brighina, Vincenzo Di Stefano

**Affiliations:** 1Neurology Unit, Department of Medicine, Surgery and Dentistry “Scuola Medica Salernitana”, University of Salerno, 84131 Salerno, Italy; claudiavinci@hotmail.it (C.V.);; 2Neurology Unit, Department of Biomedicine, Neuroscience and Advanced Diagnostics (Bi.N.D.), University of Palermo, 90127 Palermo, Italyvincenzo19689@gmail.com (V.D.S.); 3Department of Medical, Surgical and Neurological Sciences, University of Siena, 53100 Siena, Italy; federica.ginanneschi@unisi.it (F.G.);

**Keywords:** myasthenia gravis, antibodies, acetylcholine receptor (AChR), muscle-specific tyrosine kinases antibodies (MuSKs), low-density lipoprotein receptor-related protein 4 (LRP4), neuromuscular junction

## Abstract

Myasthenia gravis (MG) is an antibody-mediated neuromuscular disease affecting the neuromuscular junction. In most cases, autoantibodies can be detected in the sera of MG patients, thus aiding in diagnosis and allowing for early screening. However, there is a small proportion of patients who have no detectable auto-antibodies, a condition termed “seronegative MG” (SnMG). Several factors contribute to this, including laboratory test inaccuracies, decreased antibody production, immunosuppressive therapy, immunodeficiencies, antigen depletion, and immune-senescence. The diagnosis of SnMG is more challenging and is based on clinical features and neurophysiological tests. The early identification of these patients is needed in order to ensure early treatment and prevent complications. This narrative review aims to examine the latest updates on SnMG, defining the clinical characteristics of affected patients, diagnostic methods, management, and therapeutic scenarios.

## 1. Introduction

Myasthenia gravis (MG) is an autoimmune neuromuscular disorder in which antibodies (Abs) target specific proteins involved in the neuromuscular junction function (NMJ) [1]. These mechanisms lead to impaired NMJ transmission, resulting in fluctuating muscle weakness and increased susceptibility to fatigue [1]. The clinical presentation, treatment response, and disease mechanism differ among MG subgroups, which are classified according to the Abs pattern [2,3]. Epitope specificity and Abs characteristics play a significant role in determining disease severity, rather than their overall blood concentration levels.

The most common Abs found in MG patients are directed against the acetylcholine receptor (AChR), thus defining the classic seropositive MG phenotype. These Abs are present in about 85% of patients with generalized MG and in about 50% of patients with ocular MG [3,4,5]. On the other hand, anti-muscle-specific tyrosine kinases (MuSKs) are found in approximately 5% of patients with seronegative AChR MGA [6]. The MuSK protein is involved in the reassembly of the AChR and in the maintenance of the NMJ. In approximately 15% of patients with generalized MG and 50% of patients with ocular MG, both AChR and MuSK antibodies were absent. These patients are referred to as double-seronegative MG (dSnMG) [7,8]. Within the dSnMG group, 2–27% of patients have Abs direct to the low-density lipoprotein receptor-related protein 4 (LRP4), which plays a role in initiating the AChR cluster, defining them as triple seronegative MG (tSNMG) [7,8].

Moreover, Anti-Agrin Abs, a heparan sulfate proteoglycan released from motor nerve endings [9,10], can be detected in a minority of patients with MG, with or without AChR, MuSK, or LRP4 Abs [9,11].

Additionally, some MG patients may have muscle Abs that react with non-junctional antigens such as titin, KV1.4, and RyR [3].

Currently, radioimmunoassay (RIA) is the standard method for detecting MG antibodies, MG Abs [12]. However, in recent years, improved and highly sensitive cellular assays (CBA) or radioimmunoprecipitation assays (RIPA) have made it possible to detect anti-AChR and anti-MuSK antibodies in patients previously diagnosed with SnMG. These include low-affinity and targeting clusters of AChR Abs [12,13,14].

Furthermore, the absence of autoantibodies often delays diagnosis and the implementation of appropriate treatment, making the search for new Abs crucial. Neurophysiological techniques such as repetitive nerve stimulation (RNS) and single-fiber electromyography (SFEMG) are useful tools to achieve a correct diagnosis in SnMG. However, these techniques are not widely available in peripheral hospitals and, even in the presence of electrophysiological positivity, there is a need for a differential diagnosis with other NMJ diseases that may not benefit from immunosuppressive therapy [15,16].

A better understanding of the serological profile of MG patients has the potential to promote targeted therapeutic approaches. Significant progress has been made in recent years in the development of antigen-specific therapies that specifically target immune cells or autoantibodies involved in the autoimmune response. This review aims to highlight advances in this field, including the development of more sensitive tests for the detection of autoantibodies, the definition of clinical disease phenotypes, the sensitivity of neurophysiological methods, and the implications for correct diagnosis and early treatment.

## 2. Epidemiological and Clinical Aspects

The Abs detection is particularly crucial in determining the heterogeneous group of MG patients with different phenotypes in terms of onset, clinical course, thymic disease, and response to treatments. In the absence of anti-AChR Abs, clinically and neurophysiologically-confirmed MG patients are classified as AChR-SnMG. Anti-Musk MG and the dSNMG (negative to both anti-AchR and anti-Musk Abs), and tSNMG (negative to anti-AchR, anti-Musk, and anti-LRP4 antibodies) should be included in this group [3,4].

Although the exact causes of seronegativity remain unknown, there are considerable differences in the frequency and clinical characteristics of SnMG in relation to different ethnicities and regions. For example, among anti-AchR seronegatives, the incidence of MuSK-MG varies significantly between regions, showing an inverse correlation with geographic latitude in Europe and North America [17]. African Americans have been reported to have a significantly higher rate of anti-MuSK abs positive than Caucasians [18], compared to a higher rate of double seronegative MG (AChR/MuSK) in Europe. These ethnic and regional disparities may result from genetic predisposition and variations in environmental exposure, highlighting the need for further research in this area. Furthermore, the diagnosis of seronegative MG may require different approaches in regions with limited access to serological testing based on newly identified antigens and novel tests (i.e., anti-MuSK or anti-LRP4). The frequency and general characteristics of SnMG remain undetermined in many regions, underlining the importance of further investigations.

SnMG is often undistinguishable from other MG subtypes in terms of the distribution of strength deficit, disease severity, and response to immunosuppressive and immunomodulatory therapy (such as plasmapheresis or intravenous immunoglobulin). This suggests an underlying immune-mediated postsynaptic mechanism of the disease similar to that of the seropositive forms. However, based on the detection of other specific antibodies, different clinical MG phenotypes with different patterns of weakness distribution are identified [4].

In a recent work, [19] authors investigated the Abs profiles and clinical characteristics of 70 Korean MG patients seronegative for AChR at the conventional dosage.

Significant differences were observed between the three subgroups (anti-MuSK, pooled anti-AChR, and triple seronegative) in terms of gender, current acetylcholinesterase inhibitor use, and current MG Foundation of American (MGFA) classification. Patients with anti-Musk abs are mostly female, with a peak incidence in the fourth decade and a phenotype with prevalent involvement of the bulbar and respiratory muscles. Thymic hyperplasia and thymoma were rare in this subgroup and more than half of the patients presented myasthenic crises [20]. On the other hand, patients with pooled anti-AChR antibodies tended to have milder symptoms, often presented with the ocular form of MG, had a better prognosis with higher remission rates, and had lower seizure rates [4,8]. Two patients were positive for anti-LRP4 Abs, one with late onset generalized MG with small cell lung cancer and the other with thymoma. Patients diagnosed with dSnMG, for example, are often children or young adults and are more prone to ocular myasthenia gravis (OMG) [4,8]. They have a lower risk of developing thymoma, a type of tumor in the thymus gland, and would have a less severe disease course than HIV-positive people. However, limited research is available regarding the clinical characteristics of these patients, who test negative for both anti-AChR and anti-MuSK abs using both the radioimmunoprecipitation assay (RIPA) and cell-based assay (CBA).

Another study [8] examined a group of dSnMG (635 patients) for anti-LRP4 antibodies by the CBA method. The overall frequency of LRP4-MG in the dSnMG group (635 patients) was 18.7%, but with variations between different populations (range 7–32.7%). Patients with LRP4-MG had mild symptoms at disease onset (81% had MGFA grade I or II), with some thymic changes identified (32% hyperplasia, none with thymoma). Twenty-seven percent of patients with ocular dSnMG were positive for LRP4 antibodies. The prevalence was higher in women than in men (female/male ratio 2.5/1), with a mean onset of disease at 33.4 years for females and 41.9 for males.

A recent study conducted on a Chinese population showed that the clinical characteristics of TSnMG patients varied between different age groups. Significant findings included a bimodal distribution of the age of onset (0–9 y/o and 40–49), coexisting thymic hyperplasia, and a generally favorable prognosis. Adult patients were more likely to have generalization from ocular forms, while juvenile subjects were more likely to experience clinical relapse [21].

## 3. The Role of Neurophysiological Testing

Neurophysiology has a central role in diagnosing abnormalities of the neuromuscular transmission. Its importance becomes paramount in diagnostically challenging neuromuscular junction (NMJ) diseases, such as SnMG. There are two classic electrophysiological techniques to evaluate NMJ function: repetitive nerve stimulation (RNS) and jitter analysis in single-fiber muscle electromyography (SFEMG).

SFEMG is considered the most sensitive test when prescribed for diagnostic purposes when a NMJ disorder is suspected or must be refused. SFEMG sensitivity for diagnosing MG is usually reported between 80 and 100% in generalized MG and 93% in ocular MG [22]. Harrison et al. [23], using SFEMG, were able to reveal eight mimics of MG in 61 patients previously diagnosed with MG. These cases were predominantly seronegative for anti-AChR, anti-MusK, and anti-LRP4 antibodies, thus emphasizing the importance of SFEMG in this context.

SFEMG measures jitter, which represents the variability in the firing of two individual muscle fibers belonging to the same motor unit. Jitter is usually expressed as the mean consecutive difference of the inter-potential interval between 20 pairs of muscle fibers. Abnormal jitter has variably been defined as mean jitter exceeding 40 μs or 10% of potential pairs having block or jitter exceeding 54 μs. The muscle selection should be tailored to the clinical symptoms of subjects. This is because a normal jitter study in a weak muscle virtually excludes abnormality in neuromuscular transmission, whereas a normal study in an asymptomatic muscle neither refutes nor proves the presence of a defect in NMJ transmission. Frontalis, orbicularis oculi, and extensor digitorum communis are commonly tested muscles in SFEMG, which may be performed with voluntary contraction or with the aid of electrical stimulation.

We acknowledge that the interpretation of jitter values can be influenced by various factors, including the specific methodology employed (voluntary or stimulated) and the muscle examined. Different research groups might use slightly different protocols, electrode placements, or analysis techniques, leading to variability in the reported jitter values [24,25,26,27].

SFEMG is not a confirmatory test for the diagnosis of MG; however, it has a high negative predictive value in identifying patients without MG [24]. In this regard, SFEMG can be a useful tool for the diagnostic process in patients with typical symptoms of MG and in SnMG.

Despite the advantages of SFEMG, RNS is also performed when assessing for NMJ disorder. RNS depletes the presynaptic nerve terminal of acetylcholine, and while a normal NMJ can compensate, an abnormal one cannot. The sensitivity of RNS is around 80% in generalized MG, but can be less than 40% in ocular MG [25]. However, RNS has the highest specificity amongst neurophysiological tests for diagnosing MG [26]; hence, the combination of abnormal RNS and SFEMG showing both jitter and blocking may be highly specific for a diagnosis of MG in seronegative subjects.

RNS is typically performed by stimulating a nerve at 2–5 Hz and recording the compound muscle action potential (CMAP) from the corresponding muscle at rest and after stimulation. A 10% decrement of CMAP amplitude between the first and the fourth has traditionally been used as the cut-off for diagnosing MG. The results of RNS studies mainly depend on which nerve and muscle pair is stimulated. The choice should be made on the basis of MG type; for example, the facial muscles are relatively more affected in ocular MG. Since abnormal RNS in even one muscle can be meaningful, several muscles should be bilaterally tested in order to increase the diagnostic RNS sensitivity.

Finally, we underline that unreliable electrophysiological results are the major contributor in the overdiagnosis of MG, which is not uncommon and occurs more frequently in SnMG patients [27]. This observation emphasizes the importance of having specialized electrophysiological studies carried out at experienced centers.

## 4. Serological Diagnosis: Different Methods of Antibodies’ Detection

The “seronegative” label strongly relies not only upon how many kinds of antibodies are searched, but also upon the chosen method of ab detection, for different techniques carry variable grades of accuracy. This is especially valid for anti-AChR, since up to 85% of MG patients carry this ab, it being the most prevalent among serotypes.

To date, RIPA (Radioimmunoprecipitation Assay) is considered the “gold standard” for anti-AChR detection: it employs solubilized nicotinic AChR labeled with [125I]-α-bungarotoxin that are incubated with the patient’s serum; then, a second antihuman IgG antibody is added to precipitate the complex, and radioactivity is counted and compared to the healthy controls’ sera, being proportional to the antibodies’ concentration and therefore giving a quantitative measure of antibody titer.

Sensitivity is 80–85% in generalized MG, reaching almost 100% in thymomatous MG, but lowers in ocular MG (around 50%) and juvenile MG (especially in prepuberal onset, up to 50%), as these phenotypes are reported to carry higher rates of seronegativity [28,29].

Successful attempts were made to increase RIPA’s sensitivity by using larger volumes of test serum and decreasing background radioactivity, but these methods are not equally standardized [30]. Specificity is nearly 100%: in a 4-year prospective cohort study, only five cases of RIPA-AChR-positive non-MG cases were reported, thought to be non-pathogenetic as patients’ clinical and electrophysiological features were not consistent with MG, and their presence was not confirmed by more sensitive methods such as Live-CBA (see later) [31].

The main issues limiting the spreading of this technique are linked to the use of radioactive reagents, with related difficulties in waste handling and disposal and the brief lifespan of reagents.

In order to overcome RIA’s disadvantages, a non-radioactive, competitive Enzyme-linked immunosorbent assay (ELISA) was developed in 2006: antigen coated plate wells are covered with the patient serum; then, after several washes, a second and third biotin-labeled antibody is added. In presence of anti-AChR antibodies, the formation of this so-called “sandwich” is reduced proportionally to the amount of antibodies, providing a quantitative measure of antibody titer.

Despite initial data showing an accuracy at least as high as RIPA [32], real life results showed inferior performances with an additional 30% of false negatives and 5% of false positives compared to RIPA [33]; so, though broadly available and easy to perform, careful consideration should be given in the results’ interpretation, especially when the clinical suspect is strong.

In recent years, a novel technique called the Live Cell-based assay (CBA) has been introduced. It employs human embryonic kidney (HEK) cell cultures transfected with specific cDNAs in order to hyper-express AChRs on membrane surfaces in their native conformation; appropriate clustering of the receptors is obtained by the co-expression of the intracellular anchoring protein rapsyn. When the patient’s serum is added, antibody binding is then visualized by indirect immunofluorescence microscopy, thereby providing only a qualitative assay.

This method allows AChRs to be represented in their original folding and tridimensional structure: lower-affinity, conformation-sensitive antibodies—the so-called “Clustered-AChR Abs”—can efficaciously bind their target on the cell surface by a cross-linking mechanism, thus increasing overall sensitivity. Notably, these clustered-Abs are proved to be pathogenetic; their passive transfer in a mouse model cause the loss of post-synaptic AChR receptors.

In fact, studies report positivity ranges from 15% to 65% in RIPA-seronegative MG patients [28,30], especially regarding prepuberal-onset (up to 62%) and ocular forms (68%) of SnMG [34].

The main disadvantage that limits its use to specialized research centers is the requirement of expertise and of cell-culture facilities, with no commercial kit available.

Currently, a commercial kit of “fixed” CBA has been developed; it is composed by a biochip containing four fixed CBAs that allows the simultaneous detection of adult- and fetal-AChR and MuSK antibodies. Data regarding its accuracy are discordant: in two studies, excellent specificity (99.6% compared to RIPA) was reported, with an at least comparable if not slightly superior sensitivity compared to RIPA (4% increased, with 21% of anti-AChR detected in the RIPA-seronegative group); still, it remains less accurate than live CBA, which carries an 8% increase in sensitivity compared to RIPA [35,36]. Conversely, an Italian study comparing fixed CBA to ELISA showed a 29.5% lower sensitivity, with false negatives occurring especially in cases of low autoantibody titers, detected by ELISA, thereby suggesting the use of BIOCHIP as a screening method, whose results should be confirmed by a second technique [37]. This discordance is likely to be attributed to a low inter-rater level of agreement, for variable levels of training can affect the qualitative interpretation of the results and ambiguous staining patterns can lead to incongruent results [38]. On the other hand, the detection and measurement of serum anti-AChR antibodies performed by competitive ELISA (cELISA), indirect ELISA (iELISA), and an F-CBA, showed that cELISA has better performance analytics than iELISA and F-CBA. However, iELISA and F-CBA showed the greatest agreement [39].

Regarding anti-MuSK Abs, the second most-prevalent antibodies in MG (5–7% of subjects), analogous considerations can be made.

RIPA is the most reliable and standardized method. It allows antibody detection in 30–40% of SNMG, with a specificity that approaches 100%. A Live CBA test also exists, showing a higher sensitivity as it identifies anti-MuSK in 8% of double-seronegative patients. Of note, these subjects show a milder clinical phenotype compared to the RIPA-seropositive group [40]. Other studies point out a lower specificity compared to RIPA for 1.9% of healthy controls, and 5% of patients with other neurological disorders were found as positive on CBA [13]. Anti-LRP4 is an extremely rare antibody mainly found in ocular or mild forms of generalized MG [4]; its pathogenetic role is still in debate, for studies of the serum passive transfer in animal models are currently lacking. Detection rates in double-seronegative patients range from 2% to 50%, depending on the geographical origin, with LRP4-Ab being more frequent in non-Caucasian patients, and on the employed assay, with Live CBA being the most widely used [4]. A lower specificity must be taken into account when interpreting results: a study reported LRP4 Ab detection rates in up to 23% patients affected by motor neuron disease (MND), employing both CBA and RIPA [41]. Only a limited number of studies have provided information on the clinical characteristics of patients who tested negative for both AChR antibodies and MuSK using both the RIPA and CBA methods. These studies particularly focused on the presence of LRP4 antibodies (tSnMG) and, in a single study, Agrin antibodies (quadruple SnMG) [42].

From a practical point of view, an appropriate clinical setting, based on a suggestive symptomatology and supported by significant electrophysiological findings, is fundamental in order to avoid misleading interpretation of false positive serological results, which can also occur when employing highly specific methods. A first step should be to always test the anti-AChR abs, preferably by RIPA, but lower-sensitive methods can be used for screening if more readily available; in this second case, repeating the dosage after 6 months can help orienting the diagnosis, especially in cases of doubtful seronegativity or borderline positivity.

If the patient is “single-seronegative”, the next step is anti-MuSK testing; some clinical features, herein reported, are strongly suggestive for this serotype and should promptly lead to antibody testing. If absent, the patient is labeled as “double-seronegative”. In this case, anti-LRP4 testing, if available, could be useful mainly from a research point of view since little is known to date about clinical and prognostic features associated with this serotype. Otherwise, especially in severe forms with poor response to standard therapy, an attempt of serological relabeling could be made employing the Live CBA technique, that ultimately provides the confirmation of “double/triple seronegativity” (Figure 1).

## 5. Differential Diagnosis

It is a well-known fact that anti-AChR Ab are not present in most patients affected by ocular MG [2]. As a result, in a significant portion of patients with seronegative MG, ocular symptoms are the principal feature [43]. In these cases, the diagnosis is challenging because the tests for nicotinic agonists and neurophysiological studies might result as negative or may present conflicting interpretation [44]. On the other hand, it is also true that several neuromuscular disorders, like myopathies and congenital MG, can be mistaken for SnMG at clinical onset. Indeed, isolated eyelid ptosis is the most frequent symptom of SnMG, but it can also occur as an isolated feature at clinical onset in oculopharyngeal muscular dystrophy and mitochondrial myopathies. Moreover, neuromuscular disorders can mimic SnMG even when ptosis is associated with other clinical symptoms, including diplopia, dysphagia, and exercise intolerance.

In most cases, the characteristic feature of SNMG is that symptoms can fluctuate over days, weeks, and months. Table 1 describes sharing and distinguishing features in the most frequent neuromuscular disorders presenting with ptosis and diplopia.

As serum CPK are almost always within normal limits, it is not difficult to exclude most causes of myopathy that can present with ptosis and diplopia, such as myotonic muscular dystrophy and myositis [45,46]; however, some kind of myopathy may present with normal serum CPK: this is the case of oculopharingeal muscular dystrophy (OPMD) and Chronic Progressive External Ophtalmoparesis (CPEO) [47,48,49]. In these cases, the diagnosis can be reached through genetic testing considering the slow progression of symptoms in the presence of a family history of ptosis with a maternal (CPEO) or autosomal dominant (OPMD) inheritance pattern. Also, ophthalmoparesis points out CPEO, while dysphagia suggests OPMD [48,49]. In the absence of CPK elevation with evidence of cancer, LEMS should be excluded through neurophysiology and antibody testing [50]. A more complicated differential diagnosis is with congenital myasthenic syndromes (CMS). In fact, these disorders are mostly autosomal recessive, so a clear family history is not always available. Also, CMS share similar neurophysiological aspects with seronegative MG, adding some difficulties [15]. Of note, CMS muscular and bulbar weakness can fluctuate and get worse with heat [51,52]. Finally, some peculiar clinical features might arise suspicion of CMS: muscle fatigue, muscle weakness, muscle hypoplasia, and minor facial anomalies like low-set ears and a high-arched palate in some patients. Moreover, certain aspects of physical examination can aid in the diagnosis, such as skeletal abnormalities, cognitive impairment, and epilepsy, which are common in CMS but not in SNMG. Finally, it should be considered that unilateral ptosis and dysphagia might follow by the direct compression of cranial nerves, and direct infiltration by neoplasms [53,54] or brainstem or spinal cord infarct, but in the latter case, the acute and rapidly progressive onset points towards a pathology affecting the central nervous system (CNS) [55].

In detail, some authors underscored similarities between MG and CNS diseases such as relapsing–remitting multiple sclerosis and an ischemic stroke [55,56]. These parallels become evident through shared features, including symptom withdrawal post-treatment, highlighting the complex interplay of neurological and immunological mechanisms in these conditions.

Furthermore, several studies reported the coexistence of multiple autoimmune disorders in individuals diagnosed with MG. This raises important questions about potential shared etiological factors or underlying genetic predispositions that might contribute to the co-occurrence of these conditions [57].

## 6. Treatment

AChE inhibitors (AChE-is), particularly oral pyridostigmine, remain the primary symptomatic treatment for MG [4]. These inhibitors are generally effective for the majority of patients with anti-AchR Abs, but positive outcomes have also been observed in patients with anti-LRP4 abs [8,58] and/or anti-Agrin Abs, as well as those with triple/quadruple SN-MG [40]. Conversely, MuSK-MG patients typically do not experience improvement with pyridostigmine. Even with a low-dose treatment, these patients often report cholinergic adverse events such as muscle cramps and fasciculations. In some cases, their symptoms may worsen [59,60].

The majority of patients with MG rely on medication for immunosuppression to manage their symptoms. Individuals who experience unsatisfactory control of symptoms with AChE-is often become candidates for glucocorticoid treatment, often in combination with nonsteroidal immunosuppressive agents.

Oral prednisone or prednisolone is the primary immunotherapy used as a first-line treatment for MG. In clinical practice, a combination of immunosuppressants is typically required for most patients to prevent MG relapse and minimize the adverse effects associated with steroid use [61]. Azathioprine, mycophenolate mofetil (MMF), cyclosporine, and tacrolimus are the main agents utilized in MG treatment, with response rates of at least 80% [1]. Immunosuppression has shown significant efficacy not only in AChR antibody-positive forms of MG but also in double seronegative MG (dSNMG) and triple seronegative MG (tSNMG). However, the response in anti-MuSK MG forms is relatively less pronounced.

Romi et al. [20] examined long-term prognoses and responses to treatment between anti-AChR seropositive and dSnMG in a Norwegian population. In both groups, all patients received treatment with cholinesterase inhibitors. A percentage of seronegative (24%) and seropositive (32%) MG patients were also treated with immunosuppressive drugs, and none of them had exclusively ocular MG. The dosage and duration of immunosuppressive drug treatment were similar between seronegative and seropositive MG patients. The withdrawal of immunosuppressive drugs was successful in two seronegative (12%) and four seropositive (9%) MG patients. A small number of patients in both groups underwent plasmapheresis due to the acute deterioration of MG, regardless of planned thymus surgery (none of them had purely ocular MG). Treatment with plasmapheresis led to improvement in both seronegative and seropositive MG patients.

Thymectomy was more common in seropositive MG patients compared to seronegative MG patients (*p* < 0.001). None of the patients with purely ocular MG underwent a thymectomy. Among the thymectomized SnMG patients, two had thymic hyperplasia and one had thymic atrophy. The histological characteristics of thymic hyperplasia and atrophy were similar between seronegative and seropositive MG patients. Among the thymectomized seropositive MG patients, sixteen had thymic hyperplasia and four had thymic atrophy. The thymectomy was typically performed approximately 2.6 years after the onset of MG. No significant complications related to the surgery were reported in either group [20].

## 7. Conclusion Remarks and Future Directions

MG is a complex disease with variations in the age of onset, muscle weakness location, severity, and response to treatment. These factors can be influenced by the presence of specific Abs, which vary among different races, gender, and thymus disorders and depend on the laboratory technique used. A serological assessment is crucial for diagnosing MG and grouping patients based on their immune profile. Abs targeting postsynaptic membrane proteins are particularly important for diagnosing and managing MG, as they determine specific subtypes of the disease. Understanding the connection between the abs profile and MG pathophysiology can help the diagnosis and guide the treatment decisions. However, experts currently lack consensus on the role of these abs in predicting outcomes and their connection to thymic histology.

In clinical practice, diagnosing MG can be challenging when patients test negative in standard serological tests. This emphasizes the need to establish a consensus on clinical and electrophysiological criteria for identifying seronegative MG and its subgroups. The goal is to improve early recognition and enhance therapeutic management.

The identification of different abs including junctional and non-junctional is expanding, particularly in late-onset MG, MG thymoma, and previously considered seronegative cases. This extension improves diagnostic accuracy and provides potential markers for predicting treatment response.

Treatment for MG should be tailored to each patient, considering the specific subgroup based on the serological pattern, age of onset, thymic pathology, and muscle weakness severity. Individualized treatment and specialized follow-up are needed to adjust therapy according to symptoms and prevent exacerbations.

The ultimate objective should be to develop more specific therapies that target and suppress the autoimmune reactions causing muscle weakness and autoantibody production. However, until antigen-specific therapies become available, optimizing existing treatments for each individual with MG remains a challenge. These advancements contribute to our understanding of MG pathology and have the potential to improve patient care through faster and more accurate diagnosis and more effective disease management, ultimately enhancing the overall quality of life for MG patients.

## 8. Study Limitations

We conducted a narrative review to explore and synthesize the existing literature on SnMG, offering a more qualitative approach by examining a diverse range of sources and discussing key themes, trends, and insights. However, the absence of a predefined inclusion and exclusion criteria, systematic and structured research methodology, and quantitative data analysis can be considered a limitation of our study. Nevertheless, by providing a comprehensive overview and analysis of the available literature, we aimed to contribute to a deeper understanding of SnMG from a broader perspective.

Future prospective and longitudinal studies are needed in order to delve deeper into the realm of SnMG, contributing to a more comprehensive understanding of this condition and its intricacies.

## Figures and Tables

**Figure 1 brainsci-13-01286-f001:**
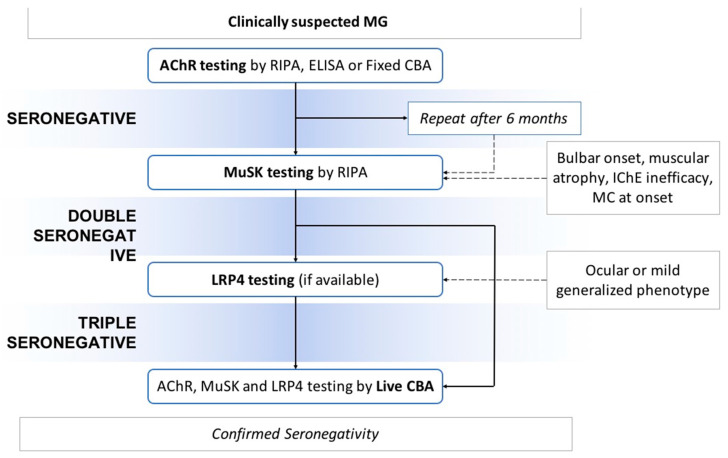
A proposed diagnostic serological algorithm in suspected MG. MG: Myasthenia Gravis, IchE: acetylcholinesterase inhibitor; CBA: cellular assays; RIPA: radioimmunoprecipitation assays; AChR: acetylcholine receptor; MuSK: muscle-specific tyrosine kinases.

**Table 1 brainsci-13-01286-t001:** SnMG mimics and chameleons with typical features and clues for differential diagnosis. OPMD, Oculo-Paringeal Muscular Dystrophy; CPEO, Chronic Progressive External Ophthalmoparesis; LEMS, Lambert-Eaton myasthenic syndrome; CMS, Congenital Myasthenia Gravis.

SnMG Mimics and Chamaleons	Etiology	Sharing Features with SnMG	Distinguishing Features	Clue for Diagnosis
Myositis	Autoimmune, iatrogenic, Viral infectious, idiopathic	Ptosis, diplopia, dysphagia, response to steroids and immunosuppressants	Proximal weakness of upper and lower limbs	Serum CPK, EMG (myopathic features, pseudomyotonic discharges), Normal RNS. Muscle biopsy (myopathic features).
Myotonic muscular dystrophies	DMPK gene mutation (19q13.3); CNBP (ZNF9) gene mutation (3q21).	Ptosis, diplopia, dysphagia	Myotonia, adult onset; proximal weakness of upper and lower limbs with early footdrop, multisystemic involvement (diabetes, cataract, cognitive impairment, baldness). Autosomal dominant inheritance	Serum CPK, EMG (myopathic features, myotonic discharges), Normal RNS. Muscle biopsy (myopathic features). CTG repeat expansion on the *DMPK* gene.
OPMD	PABPN1 gene mutation (14q11.2)	Ptosis, dysphagia	Onset in the sixth decade. Proximal weakness of upper limbs. Autosomal dominant inheritance	Serum CPK, EMG (myopathic features), Normal RNS. Muscle biopsy (myopathic features). Trinucleotide expansion of the *PABPN1* gene.
CPEO	Mitocondrial DNA mutations (ANT1, POLG, POLG2 and PEO1 genes)	Ptosis	Onset in the fifth-sixth decade, external ophtalmoparesis. Maternal inheritance.	Serum CPK, EMG (myopathic features), Normal RNS. Muscle biopsy (myopathic features). Mithocondrial DNA sequencing.
LEMS	Autoimmune, paraneoplastic, antibodies against voltage-gated calcium channels (VGCC) on presynaptic nerve terminal.	Ptosis, diplopia, dysphagia, response to steroids and immunosuppressants	Weakness of lower limbs, cramps associated with small cell lung carcinoma or thymoma, amelioration after exercise	Increased reflexes after exercise; increasing pattern at high frequency RNS; anti-presynaptic P/Q-type voltage-gated calcium channels antibodies in 60% of cases.
CMS	CHRNE, APSN, CHAT, COLQ, and DOK7 genes mutations.	Ptosis, diplopia	Juvenile onset; weakness of upper and lower limbs, in some patients low-set ears, skeletal abnormalities, cognitive impairment, epilepsy and a high-arched palate	Normal EMG with altered RNS. Genetic testing.
Horner’s syndrome	Middle ear infection to a carotid artery dissection or apical chest tumor	ptosis	Anisocoria with smaller pupil on the affected side	MRI of the brain, virus and bacteria. Carotid doppler ultrasound.
Third cranial nerve palsy	Extraorbital: diabetes, pituitary apoplexy, aneurysm, or carotid-cavernous fistula. Intraorbital: Trauma, tumors, and Tolosa-Hunt syndrome.	Unilateral ptosis, dyplopia	No fluctuation, diabetes, mechanic source of compression, herpes virus	MRI of the brain, virus, and bacteria.

## Data Availability

Data are available from the corresponding author upon a reasonable request.

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
