# Peer review of "Diagnosis and Management of Seronegative Myasthenia Gravis: Lights and Shadows"

_brainsci, 2023, doi:10.3390/brainsci13091286_

Round 1

Reviewer 1 Report

I read the paper with great interest and also must say that it was also helpfull for my knowledge as well. I think this is a good overlooking for this subject and gives simple but important highlights especially for the serological view for seronegative MG. also the brief information given about the methodology of serological methods was usefull. my only recommendation would be about the jitter values given because as you know jitter cut off lines differ due to methdology(whether stimulated SFEMG with two needles or the standart neeedle electrode used while muscle contraction is the chosen method) and the muscle examined. but the given values are all acceptable for all types off methodology. so I enjoyed wh,le reading and so thanks to authors and the editorial board for giving me this chance. 

Author Response

We thank the reviewer.Your kind words regarding the quality of our work are greatly appreciated . We modify the details you suggested in the neurophysiology section.

Reviewer 2 Report

The paper titled "Seronegative myasthenia gravis: lights and shadows" is a narrative review aiming to explore the latest advancements in seronegative MG by defining the clinical characteristics of affected patients, diagnostic methods, and management. This review is very interesting and has the following main strengths: 1) the subject is intriguing and not well researched; 2) the review thoroughly presents neurophysiological and serological diagnosis. In Table 1, the authors focus on the differential diagnosis related to ocular MG. There are many studies indicating similarity, especially in the early stages of the disease, with relapsing-remitting multiple sclerosis and ischemic stroke due to symptom withdrawal after treatment. Moreover, there are studies suggesting the coexistence of several autoimmune diseases in individuals with myasthenia. Additionally, MG patients with no detectable antibodies probably face the highest risk of inadequate MG treatment (please discuss). The title of the paper is very interesting, but it could be more specific about the scope of the review. The language of the paper is highly readable and clear. A limitation of the study is the absence of a systematic data review.

Author Response

We thank the rewiever for taking the time to review our manuscript and for your invaluable feedback. Your kind words regarding the quality of our work are greatly appreciated.

As you adviced, we have thoroughly discussed the concept of differential diagnoses with other autoimmune disorders in the related text section (red marked). This addition has indeed enriched the content and enhanced the overall comprehensiveness of the manuscript. We believe that by addressing this aspect more explicitly, we can provide a more well-rounded understanding of the subject matter.

Additionally, your advice to reconsider the title was on point. We have made the necessary changes to the title, focusing it more precisely on the key aspects highlighted in your review. (red marked) This adjustment not only reflects your insightful input but also refines the manuscript's alignment with its central themes.

Regarding the concern you raised about the absence of a systematic review of the data, we completely understand the significance of this perspective. While it is currently beyond the scope of this work, we have acknowledged this limitation in the manuscript .(red marked).

Reviewer 3 Report

The authors analyse a specific group of patients with myasthenia gravis, the so-called 'seronegative MG' (SnMG), and highlight the predisposing factors. They also discuss the diagnostic difficulties in this group of patients.

The article is well structured. It covers introduction, epidemiological and clinical aspects, role of neurophysiological tests, serological diagnosis, different methods of antibody detection, differential diagnosis, treatment and conclusions.

The methodology is not described. Addition please.

Figure is well presented.

A column with information on the aetiology of the disease, including congenital, autoimmune, etc., is missing from Table 1.

The conclusions are in line with the evidence and arguments presented; they are an answer to the research question. The article is well written and is easy to follow and understand.

none

Author Response

Thank you for your constructive feedback, which has played an instrumental role in elevating the quality and relevance of our manuscript. We performed a narrative review, so we have not included the research methods as in the case of a systematic review. We recognize this aspect as a limitation of the study and we report it in the text (section study limitations, red marked).

As you rightly suggested, we included in table 1 a column on the etiology of all disease reported.

Round 2

Reviewer 2 Report

Thank you for the opportunity for a reevaluation. The authors have considered the suggested recommendations.